# T-Cell Gene Therapy in Cancer Immunotherapy: Why It Is No Longer Just CARs on The Road

**DOI:** 10.3390/cells9071588

**Published:** 2020-06-30

**Authors:** Michael D. Crowther, Inge Marie Svane, Özcan Met

**Affiliations:** 1Department of Oncology, National Center for Cancer Immune Therapy (CCIT-DK), Copenhagen University Hospital Herlev, 2730 Herlev, Denmark; Inge.Marie.Svane@regionh.dk; 2Department of Immunology and Microbiology, Faculty of Health and Medical Sciences, University of Copenhagen, 2200 Copenhagen, Denmark

**Keywords:** CAR-T, TCR-T, T-cells, cancer, immunotherapy

## Abstract

T-cells have a natural ability to fight cancer cells in the tumour microenvironment. Due to thymic selection and tissue-driven immunomodulation, these cancer-fighting T-cells are generally low in number and exhausted. One way to overcome these issues is to genetically alter T-cells to improve their effectiveness. This process can involve introducing a receptor that has high affinity for a tumour antigen, with two promising candidates known as chimeric-antigen receptors (CARs), or T-cell receptors (TCRs) with high tumour specificity. This review focuses on the editing of immune cells to introduce such novel receptors to improve immune responses to cancer. These new receptors redirect T-cells innate killing abilities to the appropriate target on cancer cells. CARs are modified receptors that recognise whole proteins on the surface of cancer cells. They have been shown to be very effective in haematological malignancies but have limited documented efficacy in solid cancers. TCRs recognise internal antigens and therefore enable targeting of a much wider range of antigens. TCRs require major histocompatibility complex (MHC) restriction but novel TCRs may have broader antigen recognition. Moreover, there are multiple cell types which can be used as targets to improve the “off-the-shelf” capabilities of these genetic engineering methods.

## 1. Introduction

T-cells are an important immune cell capable of recognising cancers over healthy cells. The number of cancer specific T-cells in the body however is low, due to thymic selection. Furthermore, cancer specific T-cells can be turned off due to tumour-expressed immunomodulatory proteins such as PD-L1, or through dampening of the immune response due to regulatory cells [1,2,3,4]. It is possible to overcome these issues using immune checkpoint inhibitors, ex vivo expansion and adoptive cell therapy (ACT) with tumour-infiltrating lymphocytes (TILs) or genetically altering T-cells to improve their effectiveness [5,6,7] as reviewed in [8]. While isolation and expansion of TILs from melanomas is relatively straight forward, it is more difficult to obtain clinical efficacy in other solid cancers. Furthermore, while TILs are enriched for tumour-specific T-cells, they still can be extremely low in number, and adoptive transfer of TILs may result in the transfer of non-tumour reactive TILs due to lack of in vitro enrichment. A common method to improve the response to cancer is by introducing a receptor that has high affinity for a tumour antigen; namely chimeric-antigen receptors (CARs) and T-cell receptors (TCRs) (forming CAR-T cells and TCR-T cells, respectively) with high tumour specificity (Figure 1). The introduction of such receptors redirects the T-cells innate killing abilities to the right target, in this case, the cancer cells.

## 2. When CARs Become TRUCKs

The first CARs began as a single protein chain, where the extracellular component is formed of an antibody able to target CD19, joined to an intracellular domain derived from the CD3ζ signalling chain to facilitate signalling in a TCR-like manner (Figure 2) [9]. Additional engineering steps led to second generation CARs which included a costimulatory signalling domain from CD28, or the addition of the 4-1BB (CD137) signalling domain, resulting in a stronger signal cascade to lead to more effective CAR-T-cells that are now the focus of clinical trials (Figure 2) [10]. Current third generation CARs have combined these co-stimulatory signal domains (Figure 2) to further enhance efficacy. It is also possible to design CARs to instead include a TCR construct fused to a CAR signalling domain, which is advantageous as it allows intracellular targeting of antigens through the major histocompatibility complex (MHC), while maintaining the potent downstream signalling of conventional CAR constructs [11]. The main disadvantage of this type of construct is the requirement of patient-specific MHC allele matching.

The latest iteration of CAR-T cells are known as “armoured CARs”, carrying a transgenic payload giving them the name “T-cells Redirected for Universal Cytokine Killing” (TRUCKs) [12]. These genetic constructs lead to the production of cytokines directly at the targeted tissue, in addition to their direct cancer recognition (Figure 2). The pro-inflammatory cytokines, such as IL)-12, IL-2 and IL-5, aim to attract a second wave of immune cells like natural killer (NK) cells, to clear cancer cells that may invisible to the CAR-T [12]. Importantly, cytokines such as IL-12 have very potent pleotropic effects, and as such cannot be given in therapeutic doses systemically due to toxicity [13]. Therefore, targeted accumulation at the site of disease provides a very attractive alternative.

CAR-T cells have so far been shown to be most effective in CD19 positive haematological malignancies. Initial trials in patients showed remarkable anti-tumour reactivity in chronic lymphocytic leukaemia (CLL) [14], acute lymphocytic leukaemia (ALL) [15] and non-Hodgkin lymphoma (NHL) [16], using CAR-T cells targeting CD19. Indeed, patients have shown complete remission rates of 70–93% [17]. Furthermore, Tisagenlecleucel (Kymriah) is now approved to treat relapsed ALL [18], while axicabtagene ciloleucel (Yescarta) is approved to treat relapsed diffuse large B-cell lymphoma [19]. 

Presently, there are 775 CAR-related clinical trials in the clinical trials database, with 66 completed and 495 active or recruiting [20], targeting 64 different antigens [21]. The remaining 214 are not yet recruiting, suspended, terminated, withdrawn, or unknown status. While currently only approved for use in haematological malignancies, 33 of the 64 targets are exclusively targeting solid cancers, with an additional 8 targeting both haematological and solid cancers. However, efficacy against solid cancers is still unknown, so the results of these clinical trials will be invaluable. There is evidence in some solid cancers of CAR immune infiltrations at tumour sites including a phase I/II trial with HER2-targeting CARs in sarcoma [22], a preclinical study of WT4-targeting CARs in renal cell carcinoma [23] and a xenogeneic model targeting glypican-1 in multiple solid tumours [24], a potential sign that CARs may be beneficial even in some solid cancers. Indeed, the development of TRUCKs (Figure 2) is one method to improve solid tumour efficacy, by site-directed release of cytokine in order to improve survival of CAR TILs by changing the tumour microenvironment [25]. Solid tumour CARs may also be improved by altering the co-stimulatory domain (Figure 2) to induce longer-lived T-cells in vivo—as with T-cells utilising the 4-1BB domain [26]. Combination therapy of CAR-T cells with immune checkpoint inhibitors is also being explored through various clinical trials, as reviewed in [27].

## 3. Loss of CAR Antigen

While CARs have been proven to be very effective in multiple B-cell malignancies, there have been cases of resistance and relapse in patients. Mechanisms of resistance include mutations in the antigenic target resulting in loss of recognition by the CAR-T cell, mostly due to the truncation of the protein and therefore loss of surface expression of the antigen [28]. One unique case of remission occurred when a single malignant B-cell was accidently transduced with the CAR, enabling cis binding of the expressed CAR to the CD19 antigen, masking the CD19 epitope to CAR-T cells and therefore resistance to the CAR therapy [29]. To overcome mechanisms of resistance, it is possible to target a secondary molecule, such as CD22 or CD20 which are also often expressed on B-cell malignancies [30,31,32]. This can be done concurrently with anti-CD19 CARs, or as a secondary treatment post-relapse [33]. Indeed, one 8-year old patient received sequential treatments of anti-CD19 CAR, followed by anti-CD22 CAR resulting in partial response, subsequently followed by anti-CD20 CAR which cleared the cancer to complete remission [34]. Rather than giving each of these treatments sequentially, a recent CAR construct was designed to transduce donor T-cells with CD19, CD20 and CD22 targeting CARs in the same cell [35], thus cells that lose expression of one of the lineage markers would still be cleared by the adoptively transferred CARs. 

It is also possible to introduce a CAR specific to a Fcγ-receptor (FcγR), enabling restriction of CARs to an FcγR tagged tumour-targeting antibody [36,37]. This is advantageous over conventional CAR-T cells as the specificity of the CAR can be changed simply by changing the antibody. Furthermore, multiple tumour-targeting antibodies can be used at the same time, enabling the CAR-T cell to target multiple tumour antigens. FcγR CAR-T-cells can also have their toxicities managed through removal of the tumour targeting antibody, mitigating the effects of cytokine release syndrome (CRS). A similar methodology can be employed with an affinity-enhanced monomeric streptavidin 2 (mSA2) biotin-binding domain able to bind to biotin-tagged antibodies, enabling modular targeting of cancers that bind the antibody [38].

## 4. TCR-T, the Place to Be?

While CARs target surface tumour antigens with an antibody-like moiety, major histocompatibility complex restricted antigens offer a source of antigens that are both intracellular and secreted. This greatly increases the potential range of targets that can be recognised by genetically altered T-cells. One method being explored targets MHC-presented peptides using a CAR, for example recognition of α-fetoprotein peptide bound to human leukocyte antigen (HLA)-A0201 [39]. α-fetoprotein is a known cancer-associated protein but is only expressed intracellularly or secreted making it undruggable from the perspective of conventional CARs. Therefore, using the feature of MHC presentation, this has become a viable line of research. Furthermore, as TCRs are natural receptors for peptide-MHC, these remain ready to be exploited in the form of the introduction of a cancer-specific TCR into the peripheral T-cell pool analogous to CARs. Regarding antigens for these TCRs to target, there are generally two types of peptides: Those arising from mutations that differ from wild type antigens (that are not recognised by the immune system through self-tolerance mechanisms) known as neo-antigens, and those that result from over-expression or aberrantly expressed genes—known as tumour-associated antigens (TAA). Regarding the former, many avenues of research are now focusing on such a personalised treatment, due to the requirement of matching the TCR to the correct HLA, in addition to identification of cancer-specific mutations. 

There are multiple ways to identify neo-antigens, and the TCRs that recognise them. One common way is through sequencing of a patient’s cancer cells to identify cancer-specific mutations that can be targeted, which are processed through an MHC-binding prediction to enrich the antigens for those that are more likely to be presented to T-cells [40], test these using HLA-tetramers [41], finally identifying TCRs that target these antigens through clonal T-cell analyses or single cell sequencing [42,43,44]. TCR-T cells can then be synthesised and adoptively transferred to the patient. This of course greatly increases the lead time therapy; however, it is potentially advantageous in that the TCR-T have confirmed reactivity to the tumour and will be 100% matched to the patient HLA. Neoantigens-reactive T-cells are also appealing as they are specific to cancer mutations and, therefore, less likely to react to non-cancerous somatic cells. 

On the other hand, it is possible to target shared antigens across cancer and patients, which is advantageous over neoantigen reactive TCRs as TAA are often shared between individuals and between different cancers. As TAA often arise from proteins that are not usually expressed somatically in adults, they remain viable targets for TCR-T therapy. To use one case as an example, TCR-T cells engineered to target the common melanoma TAA NY-ESO-1 have been produced [45]. The responses to this TCR-T have been encouraging, with clinical benefit seen in 80% of patients.

A further opportunity to develop broad, population wide TCR-T cells is to use HLA-agnostic TCRs. This overcomes the issue of patient-specific unique MHC alleles that restrict their tumour antigens. Indeed, an exciting new avenue of research is to use the recently discovered MR1-restricted TCR known as MC.7.G5 [46]. This TCR recognises cancer regardless of tissue origin, and as MR1 is monomorphic in the population, this TCR has potential to be used as an “off-the-shelf” TCR that can be used in many patients without the caveat of MHC-restriction, analogous to CAR-T-cells. Importantly, while this TCR showed the killing of both autologous and non-autologous cancer cells when the TCR was transferred to the T-cells of a cancer patient, it remained inert to healthy cells, demonstrating its potential in cancer treatment. Other non-MHC-restricted TCRs also exist, such as anti-cancer γδ TCRs, which when transduced to donor T-cells have been shown to redirect non-cancer specific T-cells to kill cancer cells through recognition of metabolites of the mevalonate pathway that is upregulated in cancers [47]. There are also additional αβ TCRs that are not restricted by MHC but other invariant molecules, such as CD1c [48], offering further opportunities to explore TCR-T cells using these unconventional T-cells. 

A potential source of neoantigen, TAA and HLA-agnostic reactive TCRs are in the tumour microenvironment itself; it has been well documented that tumour-infiltrating lymphocytes are enriched for tumour-reactive T-cells [44,49,50,51], as evidenced by the success of TIL-based cancer immunotherapy. As TILs are often exhausted and may have limited in vivo proliferative ability once transferred back into the patient [52,53], genetically engineering T-cells with a more naïve phenotype could increase the activatory potential, compared to unedited TILs. Furthermore, use of TCR-T cells over CAR-T cells has the advantage of the potential for epitope spreading, where T-cells with different antigen specificities to the adoptively transferred targeted antigen are also activated [54].

## 5. Examples of TCR-T

TCR-T cell therapies have been slower to reach clinical approval states for treatment of cancers. However, there have been studies that show the effectiveness of such treatments, leading to over 70 clinical trials involving TCR-T therapy (Table 1) [20]. These numbers indicate a real interest in moving forward with TCR-T as a viable therapy. One promising TCR-T target is the NY-ESO-1^c125^ TCR targeting a NY-ESO-1/LAGE-1a derived peptide presented on HLA-A2, in patients with metastatic synovial sarcoma [55]. Anti-tumour responses were seen in 50% of patients, which is commendable considering metastatic synovial sarcoma is a solid cancer that is incurable with standard therapy. Furthermore, NY-ESO-1^c125^ T-cells persisted for at least 6 months in responders and showed effector memory phenotypes. NY-ESO-1 targeting TCR-T cells have also been shown to be effective in both melanoma and synovial sarcoma, with 3-year survival rates of 33% and 38%, respectively [56]. 

T-cells have also been engineered to express a TCR targeting the melanoma associated antigen gp100, together with a CAR targeting CSPG4 antigen, which has shown in vitro cytokine release and cytotoxicity [57]. These multi-target TCR-T cells have yet to be tested in animal models or humans, however. Interestingly, using multiple targets on the same cell could be exploited in the use of “logic gates”—whereby signalling domains are split across two or even three receptors, so only cells that possess both receptor targets activate the T-cell [58].

## 6. TCR-T Limitations

A confounding factor of TCR-based genetic engineering is the issue of TCR chain cross pairing. This is due to the fact TCRs exist as heterodimers and introduced TCR-α chains can pair with the endogenous TCR-β and vice versa. This is potentially hazardous as these mis-paired TCRs could harbour harmful neoreactivities [59]. There are however multiple methods to overcome mispairing issues. TCRs can be generated into a single chain CAR-like construct to prevent cross pairing, however such a construct can result in 10–100× less sensitivity for their cognate antigen [60]. However another study found the use of only the variable region of the beta chain which resulted in no loss of TCR sensitivity [61]. It is also possible to introduce a novel TCR while simultaneously knocking out the endogenous TCR. This can be achieved in two ways; either through co-transfer of a CRISPR lentivirus targeting the endogenous TCR-β chain [47] or through targeting the TCR gene insertion at both the TCR-α and TCR-β locus resulting in direct TCR gene replacement, allowing physiological expression of the engineered TCR [62]. The latter is achieved through non-viral delivery of DNA and therefore, arguably, reduces the time and expense associated with viral-based delivery of genes. The added benefit of knocking out the endogenous TCR is reduced competition for TCR co-receptors such as CD3 [63]. In a recent clinical trial, T-cells engineered with an NY-ESO-1 targeting TCR in addition to CRISPR knockout of both the endogenous TCR and PD-1, with edited T-cell detectable nine months post-transfusion, indicating viability of CRISPR edited, tumour-specific engineered TCR-T cells [64]. 

## 7. Target Cells

An exciting avenue of research is to introduce cancer-reactive TCRs using αβ-TCRs into γδ T-cells; as γδ T-cells express a different TCR isoform they cannot mis-pair introduced TCRs with endogenous. This is further advantageous in that γδ T-cells have been shown to have some natural anticancer reactivity [43,65,66,67] in addition to a range of other features not known to conventional αβ T-cells, such as antigen presentation [68]. 

Another potential target cell line for both CAR- and TCR-constructs is NK cells; as they do not possess an endogenous TCR there is no possibility of cross pairing of TCR chains. Furthermore, NK cells are able to both directly kill cells and secrete cytokines. There are some concerns that NK cells are poor at homing to cancer sites, but this may be overcome by using more advanced TRUCKs. As an example, one group has shown editing NK cells to express CD19-reactive CARs along with the IL-15 gene and a suicide gene could effectively clear xenograft B-cell leukaemia or lymphoma in mice models [69]. Indeed, NK-CARs have also shown efficacy in humans, with 7 out of 11 patients achieving complete remission with no development of severe adverse events such as CRS [70]. These NK-CARs expanded and persisted for at least 12 months, however other NK cell-based ACT studies have often failed to detect the presence of infused NK cells no more than a couple of weeks after infusion [71,72,73]. Macrophages have also recently been used to generate CAR-Ms, demonstrating antigen-specific phagocytosis and tumour clearance in vitro, with the added benefit of direct antigen presentation and cytokine secretion [74]. CAR-Ms persisted for at least 62 days in this immune deficient NSG mouse model, however it is difficult to assess the persistence in human cancer patients. 

## 8. When to Turn Engineered Cells On and Off

Both CAR-T and TCR-T show promise but can also have side effects; notably cytokine release syndrome and immune effector cell-associated neurotoxicity syndrome (ICANS). Currently, these are treated with steroids or anti-IL-6 therapy [75], however this could diminish the overall immune response. 

An additional caveat with engineered T-cells is on-target off-tumour specific toxicity. This occurs when a cancer target is expressed on tissues other than the cancer. An example showing this is in the case of CD133-directed CAR-T cells, as while CD133 is expressed on some leukaemia cells, it is also expressed on haemopoietic stem and progenitor cells, potentially causing severe toxicity [76]. Off-target, off-tumour toxicities can also occur, as evidenced with the MAGE-A3 TCR-T that resulted in severe cardiotoxicity and death of two patients due to TCR cross-reactivity to an antigen from Titin—which is only expressed on beating cardiomyocytes [39,40]. This study was important to show how potent TCR-T cells can be in vivo, but also highlight the importance of rigorous pre-clinical testing.

There are efforts to mitigate these side-effects from the genetic construct. One method is to introduce suicide switches into the edited cells—where an administered drug at a designated interval induces cell death of only the edited cells. This can be achieved through inclusion of druggable targets such as inducible caspase-9 which leads to CAR-T cell apoptosis upon administration of a small molecule [69]. Such suicide genes prevent some of the toxicities associated with over-stimulation of CAR-T-cells [75]. It is also possible to include the aforementioned logic gates [21,31].

## 9. Concluding Remarks

Cancer immunotherapy is now a vital part of the therapy regimen for cancer treatment. With CAR-T-cells showing outstanding results in B-cell malignancies, there is scope to expand the number of targets and nuances of the therapy to improve outcomes further. Particularly, design of novel CARs to target multiple antigens concurrently represents a promising line of research, such as larger CAR constructs so individual cells can target multiple antigens, or in the case of separate CAR-T cells adoptively transferred simultaneously. The introduction of CARs to include payloads such as cytokine secretion may also provide more potent efficacy. A major caveat with CAR therapy is cytokine release syndrome due to the high affinity antigen targeting—this can be mitigated through addition of cellular switches [21], logic gates [31] or through decoupling of direct antigen targeting through the use of FcγR CARs alongside infusion of tumour-targeting antibodies [38]. This enables removal of antigen target, or dose escalation in a much more controlled manner than in direct CAR-target recognition.

Unfortunately, the evidence of effective CAR-T-cell therapy in solid tumours remains limited. While some studies have shown effects in some solid tumours, there is still more research to be done. Indeed, there is difficulty in finding solid tumour antigens due to the lack of lineage specific markers as in the case of CD19/20/22 targeting CARs. As TCRs offer targeting of tumour specific neoantigens, use of TCR-T cell could be utilised for more specific solid tumour targets. The main caveat associated with TCR-T therapy is the requirement of MHC restriction, requiring identification of patient specific antigen receptors in contrast to the broad targets of CARs. However, there are example cases of TCRs targeting common HLA alleles such NY-ESO-1 mutated antigens on HLA-A0201 [77], or TCR targeting ubiquitous molecules such as tumour-associated MR1 [46].

Changing the recognition target of T-cells is not the only mechanism of gene editing in T-cells. As TILs are already a viable treatment option in solid cancers, particularly melanoma, it is feasible to try to improve responses in treatment resistant patient. It is possible, for example, to use CRISPR to knockout T-cell exhaustion markers that may be regulating the activity of adoptively transferred T-cells; examples of which include PD-1 knockout [78,79], DGK) knockout to increase TCR activation [80], or pooled gene knock-ins to improve in vivo T-cell fitness [81]. Finally, there is recent evidence of the development of universal CAR-T-cells, where CD52 was knocked out in CAR-T cells, to prevent graft-versus-host disease by preventing CAR-T-cell loss upon anti-CD52 mediated depletion of immune cells during therapy [82]. Such, off-the-shelf treatments greatly reduce cost and lead-time to treatment, paving the way for adoption of treatment as standard of care.

## Figures and Tables

**Figure 1 cells-09-01588-f001:**
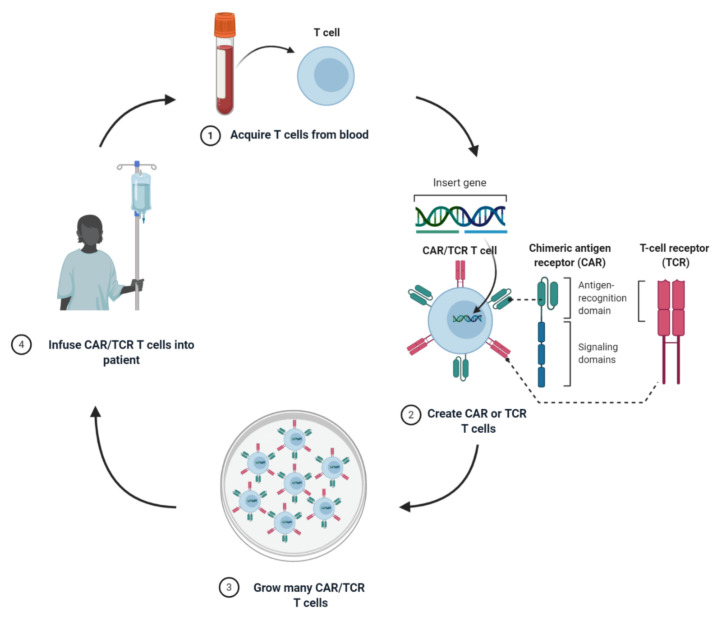
Production of chimeric-antigen receptor (CAR)/T-cell receptor (TCR) T-cells. T-cells are isolated from the blood of a cancer patient. A CAR or TCR is then introduced into the isolated T-cells through viral or non-viral delivery. CAR/TCR positive T-cells are then selected and expanded into large numbers before being transfused back into the original cancer patient.

**Figure 2 cells-09-01588-f002:**
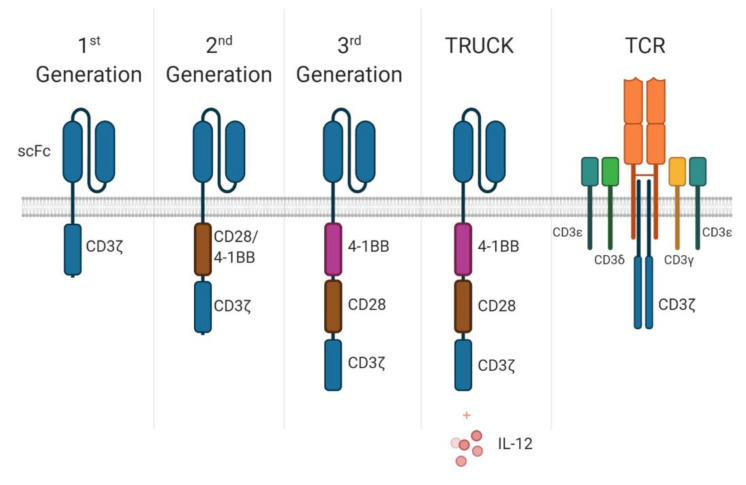
**Generational development of chimeric antigen receptors.** Chimeric antigen receptors were first developed as extracellular single-chain variable fragments (fused light and heavy chain from immunoglobulins) fused to the intracellular CD3ζ signalling domain. Then, 2nd generation CARs included the CD28 or 4-1BB co-stimulatory domains, while 3rd generation CARs included both co-stimulatory domains. The “T-cells Redirected for Universal Cytokine Killing” (TRUCKs) 4th generation also includes a cytokine payload such as IL-12. Conventional T-cell receptors on the other hand have signalling domains split across multiple proteins.

**Table 1 cells-09-01588-t001:** Clinical trials involving TCR-T therapy.

Cancer Type	Target	MHC	Clinical trial Number	Status	Year	Notes
**Cervical Intraepithelial**	HPV E7	HLA-A*02:01	NCT04411134	Phase I - Not yet recruiting	2020	
**Soft Tissue Sarcoma**	NY-ESO-1	HLA-A*02:01	NCT04318964	Phase I - Recruiting	2020	Affinity enhanced TCR
**Unresectable Hepatocellular Carcinoma**	AFP	HLA-A 02:01	NCT04368182	Phase I - Recruiting	2020	
**Solid tumour**	Tumour antigens	N/A	*NCT03891706*	Phase I - Recruiting	2019	Screen for tumour reactivity and clone TCRs
**Oropharyngeal squamous cell carcinoma**	HPV E7	HLA-A*02:01	NCT04044950	Phase II - Not yet recruiting	2019	
**Oesophagus, Hepatoma, Glioma, Gastric**	NY-ESO-1, Mesothelin, EGFRvIII and DR5	HLAA*0201	NCT03941626	Phase I/II - Recruiting	2019	CAR-T/TCR-T cells include four different tumour-specific antibodies
**HPV-Associated Oropharyngeal**	HPV E7	HLA-A 02:01	NCT04015336	Phase II - Recruiting	2019	
**Solid Tumour**	Autologous tumour antigen	N/A	NCT03970382	Phase I - Recruiting	2019	2 arms - with and without anti-PD-1 antibody
**Vulvar High-Grade Squamous Intraepithelial**	HPV E7	HLA-A*0201	NCT03937791	Phase II - Recruiting	2019	
**Head and Neck Squamous Cell Carcinoma**	HPV-16 E6	-	NCT04139057	Phase I/II - Recruiting	2019	anti-PD1 auto-secreted element
**Human Papillomavirus (HPV) 16+ Relapsed/Refractory**	HPV E7	HLA-A*02:01	NCT03912831	Phase I - Recruiting	2019	
**Nasopharyngeal Carcinoma**	LMP2	HLA-A2, HLA-A11 or HLA-A24	NCT03925896	Phase I - Recruiting	2019	
**Glioma, glioblastoma**	-	-	NCT03392545	Phase I - Recruiting	2019	
**High Grade Squamous Intraepithelial**	HPV E6	HLA-A*02:01	NCT03197025	Phase I - Completed	2019	A single patient, no response
**Pancreatic**	KRAS G12V Mutant	HLA-C*08:02	NCT04146298	Phase I/II - Recruiting	2019	Anti-PD1 adjuvant if required
**Breast**	Autologous tumour antigen	-	NCT04194190	Individual Patient Expanded Access	2019	Single patient individual access
**Prostate**	TP53	-	NCT04135092	Individual Patient Expanded Access	2019	Single Patient Expanded Access
**Hepatocellular Carcinoma**	AFP	HLA-A 02:01	NCT03971747	Phase I - Recruiting	2019	
**Non-small cell lung**	Tumour antigens	N/A	NCT03778814	Phase I - Recruiting	2018	Screen for tumour reactivity and clone TCRs
**Cervical Cancer, Head and Neck Squamous Cell Carcinoma**	HPV-E6	Not provided	NCT03578406	Phase I - Recruiting	2018	TCR-T secretion of anti-PD1
**Wide range of solid and liquid cancers**	CD19, CD22, CD33, CD38, BCMA, NY-ESO-1, c-met, Mesothelin, CEGFRvIII and DR5	HLA-A*0201	NCT03638206	Phase I/II - Recruiting	2018	
**Colorectal**	TGFβRII frameshift antigen	HLA-A*0201	NCT03431311	Phase I/II - Terminated	2018	Only 1 patient enrolled, no results given
**Fallopian Tube Carcinoma, Ovarian carcinoma, peritoneal carcinoma**	NY-ESO-1	HLA- A*02.1 and HLA-DP*04	NCT03691376	Phase I - Recruiting	2018	Melphalan pre-conditioning and TCR+ HSC transfer in addition to CD8+ TCR+ cells
**Nasopharyngeal Carcinoma**	LMP1, LMP2 and EBNA1	HLA-A*0201/2402/1101	NCT03648697	Phase II - Not yet recruiting	2018	
**Bone Sarcoma, Soft Tissue Sarcoma**	NY-ESO-1	HLA-A*02:01	NCT03462316	Phase I - Recruiting	2018	
**Solid cancers**	Patient-specific mutations	N/A	NCT03412877	Phase II - Recruiting	2018	
**Gastrointestinal, Pancreatic, Gastric, Colon, Rectal**	G12D Variant of Mutated RAS	HLA-A*11:01	NCT03745326	Phase I/II - Recruiting	2018	Murine TCR.
**Myeloid and Lymphoid Neoplasms**	Preferentially Expressed Antigen in Melanoma (PRAME)	HLA-A*0201	NCT03503968	Phase I/II - Recruiting	2018	
**Broad**	NY-ESO-1/ LAGE-1a	HLA-A*02:01, HLA-A*02:05, and/or HLA-A*02:06	NCT03709706	Phase II - Recruiting	2018	Includes a pembrolizumab combination treatment arm
**Merkel Cell Cancer**	Merkel cell polyomavirus	HLA-A*02:01	NCT03747484	Phase I/II - Recruiting	2018	Inclusion requires previous anti-PD1 treatment
**Lung Cancer, Non-small Cell, Recurrent**	NY-ESO-1	HLA-A*0201	NCT03029273	Phase I - Recruiting	2017	Affinity enhanced TCR
**Acute Leukaemia**	HA-1	HLA-A*0201+	NCT03326921	Phase I - Recruiting	2017	Relapsed or refractory patients after stem cell transplant
**Vaginal, Cervical, Anal, Penile, Oropharyngeal**	HPV-16 E6	HLA-A*02:01	NCT02280811	Phase I/II - Completed	2017	Partial response in 2/12 treated patients
**Clear Cell Renal Cell Carcinoma**	HERV-E	HLA-A*11:01	NCT03354390	Phase I - Recruiting	2017	HERV-E is an endogenous retrovirus
**Solid tumour**	MAGE-A3/A6	HLA-DPB1*04:01	NCT03139370	Phase I - Recruiting	2017	
**Gastrointestinal, Pancreatic, Gastric, Colon, Rectal**	G12V Variant of Mutated RAS	HLA-A*11:01	NCT03190941	Phase I/II - Recruiting	2017	Murine TCR.
**Synovial Sarcoma**	NY-ESO-1	(HLA)-A*02:01 or HLA-A*02:06	NCT03250325	Phase I/II - Active, not recruiting	2017	
**Cervical Intraepithelial Neoplasia, Carcinoma in Situ, Vulvar**	HPV E7	HLA-A*02:01	NCT02858310	Phase I/II - Recruiting	2016	
**Recurrent Hepatocellular Carcinoma**	Hepatitis B	Not given	NCT02719782	Phase I - Recruiting	2016	
**Hepatocellular Carcinoma**	Hepatitis B	Not given	NCT02686372	Phase I - Recruiting	2016	
**Haematological**	Cytomegalovirus (CMV)	HLA-A*0201	NCT02988258	Phase I - Suspended	2016	Suspended (Protocol being re-written to allow inclusion of more patients)
**Multiple**	NY-ESO-1	HLA-A*0201	NCT02775292	Phase I - Completed	2016	Also included NY-ESO-1 pulsed Dendritic cells adoptively transferred
**Multiple**	NY-ESO-1	HLA-A*0201	NCT02774291	Phase I - Recruiting	2016	Uses murine TCR
**Melanoma**	MART-1	HLA-A*0201	NCT02654821	Phase I/II - Active, not recruiting	2016	
**Multiple**	NY-ESO-1	HLA-A*0201	NCT02650986	Phase I/II - Recruiting	2016	Also included TGFbDNRII gene. Two treatment arms, one included chemotherapy drug decitabine
**Metastatic**	NY-ESO-1	HLA-A*0201	NCT02062359	Phase II - Terminated	2016	Study was closed due to poor accrual. TCR transduced into CD62L+ cells.
**Melanoma**	tyrosinase	HLA-A2	NCT02870244	Phase I - Recruiting	2016	
**Acute Myeloid Leukaemia**	Wilms tumour [WT]1	HLA-A*0201	NCT02770820	Phase I/II - Active, not recruiting	2016	
**Myelodysplastic Syndromes (MDS)Acute Myeloid Leukaemia (AML)**	WT1	HLA-A*02:01	NCT02550535	Phase I/II - Completed	2015	No results posted
**Multiple**	NY ESO-1	HLA-A*02:01	NCT02457650	Phase I - Unknown	2015	Verified August 2016 by Shenzhen Second People’s Hospital. Recruitment status was: Recruiting
**Thyroid Cancer**	Thyroglobulin	HLA-A*0201	NCT02390739	Phase I/II - Withdrawn	2015	No results posted
**Solid**	NY-ESO-1	HLA-A*0201	NCT02366546	Phase I - Active, not recruiting	2015	
**Non-small Cell Lung Cancer or Mesothelioma**	WT1	HLA-A*0201	NCT02408016	Phase I/II - Active, not recruiting	2015	
**Solid**	MAGE-A4	HLA-A*24:02	NCT02096614	Phase I - Unknown	2014	Verified November 2017 by Shinichi Kageyama, Mie University. Recruitment status was: Recruiting
**Breast, Cervical, Renal, Melanoma, Bladder**	MAGE-A3	HLA-A 01	NCT02153905	Phase I/II - Terminated	2014	Terminated due to slow, insufficient accrual. 1/3 Patients had PR
**Unidentified solid tumour**	NY-ESO-1	HLA-A*0201	NCT02070406	Phase I - Terminated	2014	Including DC vaccine and ipilumab treatment. Terminated due to low accrual.
**Melanoma, Meningioma, Breast, Non-Small Cell Lung, Hepatocellular**	NY-ESO-1	HLA-A*0201	NCT01967823	Phase II - Completed	2013	No results posted
**Metastatic**	MAGE-A3/12	HLA-A*0201	NCT01273181	Phase I/II - Terminated	2013	4/9 Patients had CR or PR
**AML and CML**	WT1	HLA-A*0201	NCT01621724	Phase I/II - Completed	2012	Completed, no results posted
**Multiple**	NY-ESO-1	HLA-A*0201	NCT01697527	Phase II - Active, not recruiting	2012	
**Ovarian**	NYESO-1c259	HLA A*0201, HLA-A*0205, and/or HLA-A*0206	NCT01567891	Phase I/II - Completed	2012	0/6 responses
**Melanoma**	tyrosinase	HLA-A2	NCT01586403	Phase I - Active, not recruiting	2012	
**Melanoma**	gp 100:154, MART-1 F5	HLA-A*0201	NCT00923195	Phase II - Completed	2012	Also, peptide vaccines. Progressive disease in 4/4 patients.
**Leukaemia**	WT1	HLA-A*0201	NCT01640301	Phase I/II - Active, not recruiting	2012	After allogeneic HCT
**Melanoma**	NY-ESO-1ᶜ^2^⁵⁹	HLA-A*0201	NCT01350401	Phase I/II - Terminated	2011	Terminated due to lack of enrolment.
**Melanoma**	TP53	HLA-A*0201	NCT00393029	Phase II - Completed	2011	1/9 Patients had tumour regression
**Multiple Myeloma**	NY-ESO-1c259	HLA-A*0201	NCT01352286	Phase II - Completed	2011	OS of 35.1.
**Renal, Kidney**	TNF-related apoptosis inducing ligand (TRAIL)	HLA-DR4	NCT00923390	Phase I/II - Terminated	2009	Terminated after 10 years, no results posted
**Kidney, Melanoma, Unspecified Adult Solid Tumour**	TP53	HLA-A*0201	NCT00704938	Phase II - Terminated	2008	Includes adenovirus p53 dendritic cell (DC) vaccine. Terminated due to withdrawal of support from collaborator. No CR or PR.
**Melanoma**	MART-1 F5	HLA-A*0201	NCT00706992	Phase II - Terminated	2008	<11 subjects were enrolled to each Arm. No immune responses observed.

Information sourced from ^20^.

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
