# Peer review of "T-Cell Gene Therapy in Cancer Immunotherapy: Why It Is No Longer Just CARs on The Road"

_cells, 2020, doi:10.3390/cells9071588_

Round 1

Reviewer 1 Report

Crowther et al provide a short but nice review of recent advances in CAR-T cell and TCR-T cell immunotherapies in their manuscript titled “T-cell gene therapy in cancer immunotherapy: why it is no longer just CARs on the road.” The manuscript is generally well written, but there are a number of points listed below to address in order to improve the review.

1) Abstract: In the abstract the sentence in lines 14-16 beginning with “Recent advances“ and ending with “high tumor specificity” is a bit difficult to follow and should be updated to improve readability. Consider removing the “;” and adding the CAR and TCR parts into the sentence or breaking it up into two sentences. Would also recommend altering sentence 12 to be “T-cells are generally low in number and exhausted” since there may be some exceptions to this.

2) Introduction: Line 40 is confusing due to the use of the word “reactive”. Consider updating it to “A common method to improve the response to cancer is by introducing . . .”

3) When CARs become TRUCKS: While TCR-CARs are addressed near the end of the review, it would be useful to include a sentence about the pros and cons of using high affinity TCRs rather than antibodies in CAR constructs. This could be done around line 60 as part of the Figure 2 discussion.

In lines 77-78 the 700 clinical trials are cited with 64 completed and 468 active or recruiting. What is the status of the other 160 or so?

In lines 80-86 list some of the solid tumor trials, there is no discussion on the novel strategies or mechanisms they are utilizing (just the targets). It would be useful to have more focus in the review on the current efforts (strategies/mechanisms) to overcome the lack of CAR-T cell success in treating solid tumors.

4) Loss of CAR antigen: Would recommend replacing “interesting” with “unique” in line 101 since “interesting” could be interpreted to be offensive to some since it is describing a patient remission case.

Lines 114-120 discuss the idea of CARs targeting Fcg-receptors, but the study cited (mSA2 Affinity-Enhanced Biotin-Binding CAR T Cells for Universal Tumor Targeting) is describing how the CARs can be designed with the streptavidin domain and bind to biotin on antibodies. This section will need to be updated for accuracy and to incorporate the novelty of the cited study.

5) TCR-T, the place to be?: Lines 168-169 have two grammatical issues to resolve. Replace “for” with “to” so it reads “. . TCR was transferred to the T-cells. . “. Also, “exampling” could be replaced with “demonstrating” or “providing an example of its potential in cancer treatment” and it would be easier to understand.

6) TCR-T limitations: Lines 211-215 discuss the use of single chain CAR-like constructs to prevent mispairing and how Harris et al showed 10-100x less sensitivity whereas another study (Oh et al) disputes that. It is important to point out that these are not comparing the same types of constructs. Harris et al is using an alpha-beta variable region single chain construct whereas Oh et al is using a construct with only the variable region of the beta chain. This should be updated for accuracy and clarity.

7) Target cells: As part of the discussion on the use of NK cells and macrophages with CARs, it would be important to add a sentence or two addressing their different lifespans compared to T cells and the pros and cons of that.

Author Response

Crowther et al provide a short but nice review of recent advances in CAR-T cell and TCR-T cell immunotherapies in their manuscript titled “T-cell gene therapy in cancer immunotherapy: why it is no longer just CARs on the road.” The manuscript is generally well written, but there are a number of points listed below to address in order to improve the review.

Many thanks for reviewing the manuscript and the constructive comments.

1) Abstract: In the abstract the sentence in lines 14-16 beginning with “Recent advances“ and ending with “high tumor specificity” is a bit difficult to follow and should be updated to improve readability. Consider removing the “;” and adding the CAR and TCR parts into the sentence or breaking it up into two sentences. Would also recommend altering sentence 12 to be “T-cells are generally low in number and exhausted” since there may be some exceptions to this.

This has now been updated.

2) Introduction: Line 40 is confusing due to the use of the word “reactive”. Consider updating it to “A common method to improve the response to cancer is by introducing . . .”

This has now been updated.

3) When CARs become TRUCKS: While TCR-CARs are addressed near the end of the review, it would be useful to include a sentence about the pros and cons of using high affinity TCRs rather than antibodies in CAR constructs. This could be done around line 60 as part of the Figure 2 discussion.

A new statement has been added at line 61. "It is also possible to design CARs to instead include a TCR construct fused to a CAR signalling domain, which is advantageous as it allows intracellular targeting of antigens through the MHC, while maintaining the potent downstream signalling of conventional CAR constructs [11]. The main disadvantage of this type of construct is the requirement of patient-specific MHC allele matching. "

In lines 77-78 the 700 clinical trials are cited with 64 completed and 468 active or recruiting. What is the status of the other 160 or so?

A line has been included describing the status of the others trials "The remaining 214 are not yet recruiting, suspended, terminated, withdrawn, or unknown status. ".

In lines 80-86 list some of the solid tumor trials, there is no discussion on the novel strategies or mechanisms they are utilizing (just the targets). It would be useful to have more focus in the review on the current efforts (strategies/mechanisms) to overcome the lack of CAR-T cell success in treating solid tumors.

A new paragraph has been included outlining some options for improving CAR T cells "Indeed, the development of TRUCKs (Figure 2) is one method to improve solid tumour efficacy, by site-directed release of cytokine in order to improve survival of CAR TILs by changing the tumour microenvironment [25]. Solid tumour CARs may also be improved by altering the co-stimulatory domain (Figure 2) to induce longer-lived T-cells in vivo – as with T-cells utilising the 4-1BB domain [26]. Combination therapy of CAR-T cells with immune checkpoint inhibitors is also being explored through various clinical trials, as reviewed in [27]."

4) Loss of CAR antigen: Would recommend replacing “interesting” with “unique” in line 101 since “interesting” could be interpreted to be offensive to some since it is describing a patient remission case.

This has been updated.

Lines 114-120 discuss the idea of CARs targeting Fcg-receptors, but the study cited (mSA2 Affinity-Enhanced Biotin-Binding CAR T Cells for Universal Tumor Targeting) is describing how the CARs can be designed with the streptavidin domain and bind to biotin on antibodies. This section will need to be updated for accuracy and to incorporate the novelty of the cited study.

This was due to placement of the wrong reference. Thank you to the reviewer for pointing this out. We have now included both articles in this review."A similar methodology can be employed with an affinity-enhanced monomeric streptavidin 2 (mSA2) biotin-binding domain able to bind to biotin-tagged antibodies, enabling modular targeting of cancers that bind the antibody [38]. "

5) TCR-T, the place to be?: Lines 168-169 have two grammatical issues to resolve. Replace “for” with “to” so it reads “. . TCR was transferred to the T-cells. . “. Also, “exampling” could be replaced with “demonstrating” or “providing an example of its potential in cancer treatment” and it would be easier to understand.

This has been updated.

6) TCR-T limitations: Lines 211-215 discuss the use of single chain CAR-like constructs to prevent mispairing and how Harris et al showed 10-100x less sensitivity whereas another study (Oh et al) disputes that. It is important to point out that these are not comparing the same types of constructs. Harris et al is using an alpha-beta variable region single chain construct whereas Oh et al is using a construct with only the variable region of the beta chain. This should be updated for accuracy and clarity.

This has now been updated "However another study found use of only the variable region of the beta chain resulted in no loss of TCR sensitivity [61]. "

7) Target cells: As part of the discussion on the use of NK cells and macrophages with CARs, it would be important to add a sentence or two addressing their different lifespans compared to T cells and the pros and cons of that.

New sentences have now been added to include this information "These NK-CARs expanded and persisted for at least 12 months, however other NK cell-based ACT studies have often failed to detect the presence of infused NK cells no more than a couple of weeks after infusion [71–73]Macrophages have also recently been used to generate CAR-Ms, demonstrating antigen-specific phagocytosis and tumour clearance in vitro, with the added benefit of direct antigen presentation and cytokine secretion [74]. CAR-Ms persisted for at least 62 days in this NSG mouse model, however it is difficult to assess the persistence in human cancer patients. "

Reviewer 2 Report

This is a well written mini-review of CAR T and TCR T cellular therapy for cancers.

It will be more reader-friendly if authors could provide a table to outline different new receptor approaches and their pros/cons.

Author Response

This is a well written mini-review of CAR T and TCR T cellular therapy for cancers.

Many thanks for the reviewer comments.

It will be more reader-friendly if authors could provide a table to outline different new receptor approaches and their pros/cons.

We have now included an extensive table outlining all the clinical trials involving TCR-T therapy, their targets, and their HLA.
